# Genomic C-Value Variation Analysis in Jujube (*Ziziphus jujuba* Mill.) in the Middle Yellow River Basin

**DOI:** 10.3390/plants12040858

**Published:** 2023-02-14

**Authors:** Hao Wu, Wanlong Su, Meijuan Shi, Xiaofang Xue, Haiyan Ren, Yongkang Wang, Ailing Zhao, Dengke Li, Mengjun Liu

**Affiliations:** 1Shanxi Key Laboratory of Fruit Germplasm Creation and Utilization, Institute of Fruit Trees, Agricultural University of Shanxi, Taiyuan 030031, China; 2College of Horticulture, Taigu Campus, Agricultural University of Shanxi, Jinzhong 030800, China; 3College of Horticulture, Baoding Campus, Agricultural University of Hebei, Baoding 071000, China

**Keywords:** Chinese jujube, genomic C-value, genetic diversity, flow cytometry, gene exchange

## Abstract

Chinese jujube (*Ziziphus jujuba* Mill.) originated in the Yellow River basin (YRB) of the Shanxi–Shaanxi region. The genomic C-value is a crucial indicator for plant breeding and germplasm evaluation. In this study, we used flow cytometry to determine the genomic C-values of jujube germplasms in the YRB of the Shanxi–Shaanxi region and evaluated their differences in different sub-regions. Of the 29 sub-regions, the highest and lowest variations were in Linxian and Xiaxian, respectively. The difference between jujube germplasms was highly significant (F = 14.89, *p* < 0.0001) in Linxian. Cluster analysis showed that both cluster 2 and 4 belonged to Linxian, which were clearly separated from other taxa but were cross-distributed in them. Linxian County is an important gene exchange center in the YRB of the Shanxi–Shaanxi region. Principal component analysis showed that cluster 1 had low genomic C-values and single-fruit weights and cluster 2 had high genomic C-values and vitamin C contents. The genomic C-value was correlated with single-fruit weight and vitamin C content. In addition, the genomic C-value was used to predict fruit agronomic traits, providing a reference for shortening the breeding cycle and genetic diversity-related studies of jujube germplasm.

## 1. Introduction

Chinese jujube (*Ziziphus jujuba* Mill., (2n = 2x = 24)) is a member of the genus *Ziziphus* in the family Rhamnaceae. It is a characteristic and dominant fruit tree in China. According to the ancient literature, the Chinese jujube originated in the Yellow River basin (YRB) of the Shanxi–Shaanxi region [1]. After a long evolutionary process, the area has nurtured rich resources of germplasm of Chinese jujube, such as ’Muzao’ and ‘Youzao,’’ totaling more than 100 species. Based on the results of molecular marking, Yin [2] hypothesized that Chinese jujube germplasm in the Yellow River basin (YRB) of the Shanxi–Shaanxi region has a high degree of interpopulation- and intrapopulation-variation and rich genetic diversity. Seven thousand years ago, Chinese ancestors started picking and cultivating jujube. Two thousand years ago, Chinese jujube cultivation areas were formed in northern China and spread to many countries in Asia and Europe with the development of the Silk Road [3,4]. Large-scale commercial cultivation of Chinese jujube is practiced in Korea and Iran. However, 98% of the Chinese jujube production area and more than 80% of the cultivated fruit trees of the genus jujube globally, are in China. Although jujube has a long cultivation history, research on the genetic diversity of jujube germplasm resources is limited. The in-depth investigation and utilization of germplasm resources are vital in terms of scientific and technological support for seed industry innovation and industrial development.

The genomic C-value is described as the amount of DNA in the gametes of unreplicated haploid chromosomes [5,6] It is a critical cytogenetic feature useful for delineating taxa [7,8,9,10,11,12,13]. Genomic C-values provide the necessary information for whole genome sequencing and therefore are important for studies on species evolution [14]. Flow cytometry data analysis is a reliable and suitable method for estimating the genome size of plant communities [15]. Genomic C-values are key indicators for plant breeding and germplasm evaluation [16] and using them to predict morphological indicators has become a predictive tool. Comertpay et al. [17] found that the meeting time of maize was delayed with an increasing DNA content and suggested that maize DNA content could be used as a selection criterion for flowering time. In contrast, Meagher et al. [18] found that DNA content was again negatively correlated with flower size. Angelino et al. [19] indicated a positive correlation between genomic C-value and seed quality in angiosperms. Korban et al. [20] determined the genomic C-value of 100 apple germplasm and showed that genomic C-value and stomatal length were positively correlated. Jujube has a long breeding cycle; therefore, introducing new cultivars may take a long time. However, genomic C-values can initially predict phenotypic traits to ensure that genomic C-values can be used as a basic breeding program.

The jujube germplasm originated in the YRB of the Shanxi–Shaanxi region, a rich center of genetic variation. Therefore, in this study, we analyze the correlation between genome size and important fruit traits. This study will serve as a reference for the genetic diversity of jujube germplasm resources and the screening of superior germplasm. In addition, correlation analysis and linear regression between genomic C-values and fruit agronomic traits were conducted to provide a basis for predicting jujube agronomic traits and reducing its breeding cycle.

## 2. Results

### 2.1. Variability of Genomic C-Value and Ploidy Analysis

Fluorescence intensity ratio and genomic C-value were analyzed using ‘Dongzao’ (*Ziziphus jujuba* Mill.) (444 Mb) [21] as a control and ‘Shanxi Duli’ (*Pyrus betulifolia*) (532.7 Mb) [22] as an internal standard. Figure 1 shows the fluorescence intensity ratio obtained by mixing samples of ‘Dongzao’ and ‘Shanxi Duli’ as 120,035/100,132. The estimated genomic C-value of jujube was 444.36 Mb with a coefficient of variation (CV) of 0.05%, indicating that it is possible to calculate the germplasm of jujube from different regions of the YRB of the Shanxi–Shaanxi region.

Flow cytometry was used for the genomic C-value and ploidy determination of 186 jujube germplasms. Results of the analysis showed significantly high differences in the genome sizes of different jujube germplasms in the YRB of the Shanxi–Shaanxi region (F = 10.89, *p* < 0.001). The highest genomic C-value was that of YJHMZ (0.286 pg) and the lowest was that of YCZZXSYZ (0.204 pg). The mean genomic C-value of the 186 germplasms of jujube along the YRB of the Shanxi–Shaanxi region (Figure 2) was 0.221 pg with a CV of 0.052. The mean genomic C-value of the jujube germplasm in Shanxi province was 0.224 pg with a CV of 0.055. The mean genomic C-value of jujube germplasms in the Shaanxi province was 0.219 pg with a CV of 0.045. Non-parametric tests showed significant differences between the Shanxi and Shaanxi populations (*p* < 0.001). ‘Dongzao’ is diploid (444 Mb, 0.4540 pg), and its unreplicated haploid DNA amount is 0.2270 pg; therefore, we determined that most jujube germplasm is diploid (2n = 2x = 24), based on flow cytometry fluorescence intensity analysis results, and hypothesized that QXPGZ (626.03 Mb, 0.6401 pg), TGDGHPZ (660.52 Mb, 0.6685 pg), JSNYDBZ (637.11 Mb, 0.6514 pg) and LXEBCSZ (675.16 Mb, 0.6903 pg) were presumed to be natural triploids. These jujube germplasms were verified using chromosome filming (Figure 3) and phenotypic traits (Figure 4). It was confirmed that QXPGZ, TGDGHPZ, JSNYDBZ, and LXEBCSZ were natural triploids.

Three sub-regions of the germplasm were selected for statistical analysis, and the genomic C-values and coefficients of variation differed among the regions, as shown in Figure 5. The largest coefficient of variation in jujube germplasm in the different areas of the same collection site was Linxian (CV = 0.057), indicating a richer genetic base of jujube germplasm in Linxian County. The smallest coefficient of variation was in Xiaxian County (CV = 0.006). Multiple comparisons using Dunnett’s test revealed significant differences (*p* < 0.05) between jujube germplasm from Linxian County and that from Pucheng County. The jujube germplasm of Dali county was significantly different from that of Xiaxian County (*p* < 0.05). There was a significant difference (*p* < 0.05) between the jujube germplasm from Xiaxian County and that from Pucheng County.

Genomic C-values are associated with the genetic characteristics of plants and have an essential role in plant genetic evolution [23,24]. Therefore, genomic C-values of 53 jujube germplasms from Linxian County were analyzed for variability. Results showed that the differences between the jujube germplasm of the Linxian County were highly significant (F = 14.89, *p* < 0.0001). The least significant difference (LSD) multiple comparisons are shown in Table 1. The highest genomic C-value of the jujube germplasm of Linxian County was for LXMJW-1 (0.25559 pg) and the lowest was for LXYLLZ (0.20582 pg), with a mean value of 0.2277 pg. Overall, the genetic diversity of the jujube germplasm is the richest in the YRB in the Shanxi–Shaanxi region of Linxian County.

### 2.2. Principal Component Analysis and Cluster Analysis of Genome Size and Phenotypic Traits in Jujube Germplasm

A principal component analysis (PCA) of genomic C-values and fruit traits in 89 jujube germplasms showed three clusters (Figure 6, left panel). Cluster 1 consisted of germplasms with a low single-fruit weight. Cluster 2 consisted of germplasms with high genomic C-values and vitamin C content. Cluster 3 consisted of germplasms with a high single-fruit weight and low vitamin C content (Table 2). Cluster analysis showed (Figure 6, right panel) that both cluster II and cluster IV were jujube germplasm from Linxian County, which were separated from those of other regions, probably because of the geographical environment, thus retaining a richer genetic variation. Other jujube germplasms in Linxian County are cross-distributed to other taxa, suggesting that Linxian may have diverse jujube plants that played a vital role in the genetic evolution of the jujube. PCA and cluster analysis showed that jujube germplasms in the YRB in the Shanxi–Shaanxi region did not cluster exactly according to geographic origin, probably because of frequent gene exchange and complex and diverse genetic backgrounds in various areas.

### 2.3. Correlation Analysis between Genomic C-values and Agronomic Traits of Jujube Germplasm

To investigate the association between genome size and phenotypic traits, 89 excellent germplasm representatives from different sub-regions were selected for correlation analysis. Results showed that genomic C-values were significantly and positively correlated with single-fruit weight (*p* < 0.05) and had a very high significant positive correlation with vitamin C content (*p* < 0.001) (Figure 7, left). We performed a linear regression analysis of the indicators with substantial correlations. The results showed that genomic C-values fitted significantly with single-fruit weight (R^2^ = 0.059, *p* < 0.01) and were highly significant with vitamin C content (R^2^ = 0.148, *p* < 0.0001) (Figure 7, right).

## 3. Discussion

### 3.1. Genomic C-Values and Variation of Jujube Germplasm

In this study, we used flow cytometry to determine the genome sizes of 186 jujube germplasms of Chinese and wild jujube along the YRB of the Shanxi–Shaanxi region for the first time. Genomic C-values differed among varieties within the same genus [25,26,27]. The genomic C-values determined in this study also differed among jujube germplasms along the YRB of the Shanxi–Shaanxi region, which is consistent with the results of previous studies. There are many examples of intra-species variation in genomic C-value size between populations that are geographically distant or growing in different ecological environments. However, the degree of variation is not very high. Liu et al. [28] reported the genomic C-values of mango germplasm from various regions of Yunnan province. The CV of mango germplasm in Honghe County was the highest (1.87%). Quan et al. [29] reported the genomic C-values of 180 *Arabidopsis thaliana* plants in Sweden and found a CV of only 1%. Noirot et al. [30] reported the genomic C-values of coffee germplasm from the Congo and Cameroon regions. The difference between the two regions was not significant with a CV of only 2%. Nowicka et al. [31] reported the genomic C-values for the genus *Daucus* (carrot) in the family Umbelliferae worldwide, with a CV of 3% for edible carrots, and indicated that the degree of intraspecific variation in cultivated carrots is not high. We showed that the mean genomic C-value of 186 jujube germplasms was 0.221 pg with a CV of 0.052. The CVs’ genomic C-value of jujube germplasms in the Shanxi and Shaanxi provinces were 0.055 and 0.045, respectively. Among the 29 sub-regions in the YRB of the Shanxi–Shaanxi region, the CV of jujube germplasm in Linxian County was the highest at 5.72%, which was larger than that reported in previous studies. Previous studies have shown that genomic C-values are related to the geographic environment [32,33,34,35,36,37]. In addition, Linxian County is a loess, hilly and ravine area with high elevation differences and complex topography, which may have influenced the genomic C-values of jujube germplasms. ANOVA and multiple comparisons of jujube germplasms in Linxian showed that the differences between individual jujube germplasms were highly significant (F = 14.89, *p* < 0.0001). Genomic C-values are also attracting increasing attention from plant taxonomists, as they often help to delineate different taxa and have the potential to influence taxonomic decisions [38]. The best methods to study plant taxonomy are PCA and cluster analysis [39,40]. Cluster analysis showed that jujube germplasms in Linxian and other sub-regions were interrelated and independently clustered, indicating that their genetic base in Linxian is richer and the genetic background is more complex. Therefore, Linxian County is an important center of genetic diversity for the origin of jujube. The origin and evolution of the species in the YRB of the Shanxi–Shaanxi region cannot be proved only by genomic C-values, and therefore, subsequent studies on resequencing and agronomic trait identification are required to verify the genetic structure and diversity of jujube in the YRB of the Shanxi–Shaanxi region.

### 3.2. Ploidy of Jujube Germplasm

This study identified the ploidy of jujube germplasm using flow cytometry and chromosome compression. Results showed that QXPGZ (636.06 Mb), TGDGHPZ (671.1 Mb), JSNYDBZ (647.31 Mb), and LXEBCSZ (685.98 Mb) were natural triploids, whereas the other jujube germplasms were diploids. Previous studies have illustrated that plant polyploidy promotes genome evolution as well as species diversity [41]. The natural jujube chromosomes were stable; however, four natural triploids were found in the place of origin, laterally indicating the rich genetic diversity in the YRB of the Shanxi–Shaanxi region. We used genomic C-values for data analysis to exclude the influence of polyploidy on results. However, genomic C-values remained variable in this study, probably because of the variation in the copy number of repetitive DNA sequences, which is a common factor in angiosperms. Hawkins et al. [42] analyzed genome size evolution across land plants and determined that the underlying trend in genome size is increasing, suggesting that older plant lineages have relatively small genomes. The genomic C-values of 0.2191 pg for wild jujube and 0.2248 pg for Chinese jujube determined in this study indicate that wild jujube may be an ancient species and that the Chinese jujube may have evolved slowly from it. Previous studies have also suggested that wild jujube may be the ancestor of the Chinese jujube [43,44].

### 3.3. Correlation between Genomic C-Values and Fruit Traits of Jujube

Plant phenotypes often change with genomic C-values, making genomic C-values an indispensable attribute for evaluating plant phenotypes. Wang et al. [45] found that jujube genome size was significantly correlated with fruit size, fruit length, fruit width, and single-fruit weight, similar to the results of the present study. In addition, the genomic C-value was found to be significantly correlated with vitamin C content. Aliyu et al. [17] found that the fruit weight of cashew was significantly correlated with the genomic C-value. Kadkhodaei et al. [46] found that an increase in pear genome size by 1.6 pg decreased the fruit diameter by 1 cm. Sarikhani et al. [47] found that nut and kernel weights and nut size index were predicted by genome size. In this study, linear regression analysis of two fruit traits that were significantly correlated, revealing that the genomic C-value (Y) was associated with single-fruit weight (Y = 0.64X + 213.11) and vitamin C content (Y = 5.387X + 2.011). However, the linear relationship was weaker than in previous studies. It may be caused by the greater degree of variation in jujube germplasm. Genome size varies significantly between and within species, and the causes of variation may be recombination rates, tandem repeats, the proliferation of transposable elements, and ecological factors [48], while transposable elements (TEs) may have played an important role in plant evolution [42]. The insertion of TEs affects the expression of host genes in several ways. Domínguez et al. [49] found TE insertions within or near 1 kb of genes, compared them to the same genes without TE insertions at the transcriptional level, and found that most of the genes with Ts insertions were expressed more than twice those without them. If TEs are inserted into the cis-regulatory region, they may enhance gene expression and produce phenotypic differences. For example, studies on red grapes, white grapes, and blood oranges all found that the insertion of TEs into the upstream promoters of genes can cause phenotypic alterations [50,51]. It is important to note that the distribution of TEs in the host genome is not random and they show a strong preference for specific genomic regions, a preference that both reduces damage to the host and maximizes an opportunity for TE expression [52,53]. In this study, genomic C values were found to be positively correlated with vitamin C content and single-fruit weight, probably due to the insertion of transposable elements, resulting in an increased expression of the relevant genes. In-depth investigations of the relationship between genomic C-values and phenotypic traits will be conducted, using genomic-enriched repeat sequences. Jujube has a long period of reproduction and development, and the selection and introduction of new varieties are long. Therefore, correlation and linear regression analyses were used to study the relationships between variables to predict missing data based on existing data. Thus, a correlation analysis of genomic C-values and agronomic traits can help shorten the breeding cycle.

## 4. Materials and Methods

### 4.1. Plant Material

In this study, 186 jujube germplasm resources (Appendix A) from 29 counties and cities in the YRB of the Shanxi–Shaanxi region, including Linxian, Xianfen, Yonghe, Loubei, Jishan, Xixian, Pinglu, Pingyao, Taigu, Yuzi, Yanchuan, Lintong, Yanliang, Xingxian, Qingxu, Jiaxian, Shilou, Linyi, Xiaxian, Taiyuan, Jiaocheng, Dali, Pucheng, Baode, Zhongyang, Yongji, Xi’an, Qixian, and Yuncheng, were selected as test material. They are kept in the National Horticultural Germplasm Repository–Jujube Subrepository, and the rootstock is the common wild jujube. The jujube repository is located in Taigu District, Jinzhong City, Shanxi Province (112°32′ E, 37°23′ N), at an altitude of 830 m, with an average annual temperature of 10.6 °C a frost-free period of 170–180 d, annual precipitation of 400–600 mm, sandy loam soil with a pH of 7.5, and the typical climate and ecological conditions of a loess plateau. Three trees per germplasm were collected, and each tree was collected with measurements of the different orientations of young leaves for flow cytometry.

### 4.2. Identification and Evaluation of Important Fruit Traits

A total of 30 to 50 representative fruits from each germplasm were selected for measurement. Single-fruit and kernel weights were measured using an analytical balance, and fruit length and fruit width were measured using vernier calipers. Soluble solids, soluble sugars, titratable acids, and vitamin C were determined using the refractometry, redox titration, indicator titration, and 2,6-dichloroindophenol titration methods, respectively. Fresh edible rate was determined as follows: (single-fruit weight − kernel weight)/single-fruit weight × 100%.

### 4.3. Genome Size Assessment

Jujube genome size was estimated according to Wang et al. method [54]. Fresh young-leaf tissue of 1–2 cm was placed in a pre-cooled Petri dish at 4 °C. A total of 1 mL of woody plant buffer [55] buffer was added to the Petri dish and chopped with a razor blade. The fresh shoot-leaf tissue was immersed in the buffer throughout the cutting process. The mixture was allowed to stand for 5 min and filtered through a flow cell-specific cell sieve into a 1.5-mL centrifuge tube, incubated for 5 min at 4 °C, and centrifuged at 1100 rpm for 5 min at 4 °C. Approximately 30 µL of supernatant was removed and 30 µL of propidium iodide dye (1 mg/L) and 1 µL of RNase solution (10 mg/L) were added. The samples were placed in a refrigerator at 4 °C for 30 min to avoid light staining. Cell suspensions were prepared by mixing at least three individual leaves selected from each jujube germplasm and passed through three technical replicates. Each sample was assayed using a BD Accuri™ C6 plus Flow Cytometer System (BD Biosciences, San Jose, CA, USA) with a minimum of 10,000 particles collected, and each replicate was assayed 3–5 times. Fluorescence intensity was detected using the FL2 fluorescence channel. CV was controlled to < 5% using BD CSampler™ Plus software to ensure accurate readings [56].

### 4.4. Chromosome Count

According to Chen et al. [57], the method for chromosome preparation was improved. Young jujube leaves were pretreated in 0.002 M 8-hydroxyquinoline solution for 2–3 h, placed in distilled water for 30 min with hypotonicity and fixed using a fixative solution for 24 h. The leaves were then dissociated with a 1:1 mixture of 5% cellulase and 5% pectinase for 3–5 h, followed by the addition of distilled water with hypotonicity for 30 min. The treated young jujube leaves were evenly coated on slides and stained for 30 min using Giemsa staining solution (2–5%).

### 4.5. Statistical Analyses

The genome size measure is the cellular C-value; 1 pg = 978 Mb [58]. The genome size of the species to be tested = (fluorescence intensity of the species to be tested/fluorescence intensity of the standard internal species) × fluorescence intensity of the standard internal species. Mean, standard deviation, and CV were calculated using Excel 2019. TheShapiro.test, Wilcox.test, and Levene’s test (in the car package), and Fligner.test functions in the R stats package were used for homogeneity of variance, normality distribution test and non-parametric tests, respectively. The LSD.test in the R agricolae package was used for multiple comparisons. The corrplot package was used for correlation analysis. The Rggplot2 package was used to produce box plots and for linear regression. One-way ANOVA as well as Dunnett’s multiple comparisons tests were performed using SPSS 35.0. ggtree and factoextra packages in R were used for cluster and principal component analyses.

## 5. Conclusions

This study showed differences in genomic C-values of jujube germplasm in the YRB of the Shanxi–Shaanxi region. Principal component and cluster analyses showed independently clustered individual jujube germplasm in Linxian County Most of the jujube germplasms were cross-linked with those of other sub-regions, probably because of the frequent gene exchange in the YRB of the Shanxi–Shaanxi region. In conclusion, Linxian County may be an essential center for gene exchange and a rich center of genetic variation in jujube. Determining genomic C-values to predict fruit agronomic traits provides a reference for studies related to the jujube germplasm’s genetic diversity, fruit evaluation, and identification.

## Figures and Tables

**Figure 1 plants-12-00858-f001:**
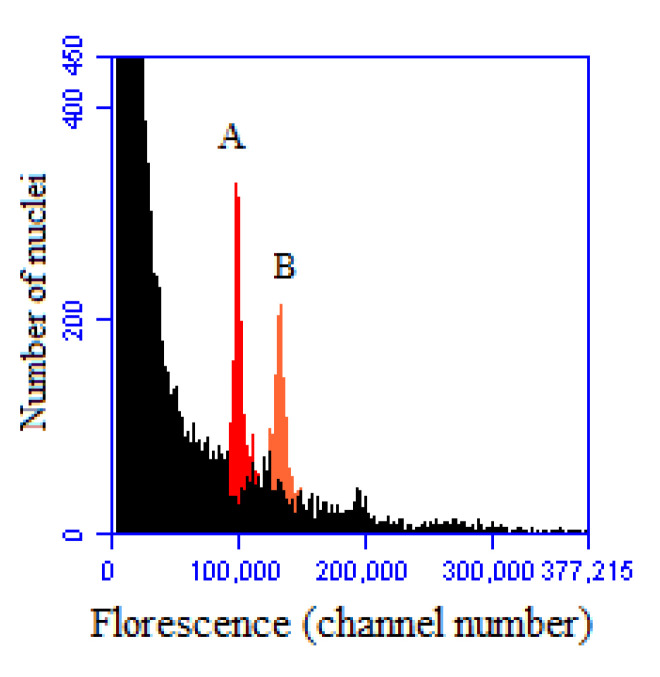
Histogram of measurements of the mixed sample. A: ‘Dongzao’; B: ‘Shanxi Duli.’.

**Figure 2 plants-12-00858-f002:**
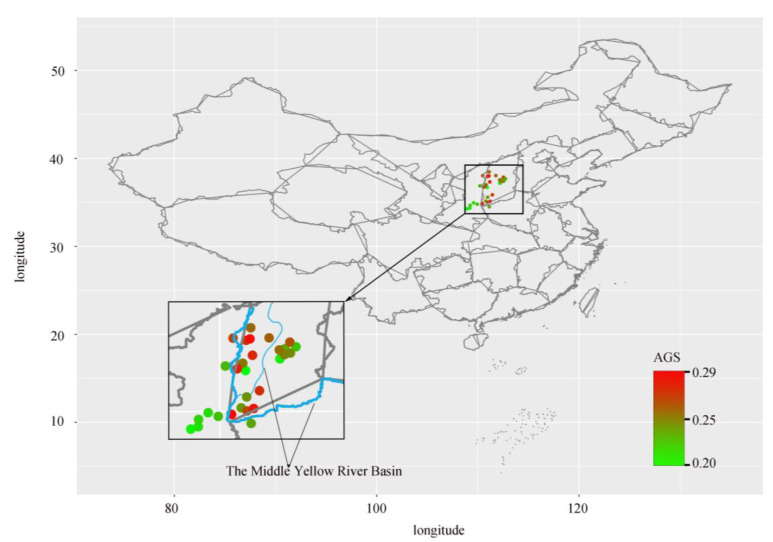
Map of average genomic C-values of jujube germplasm in the middle Yellow River basin by sub-region. AGS: Average genomic C-value. The points represent the geographical coordinates of each sub-region of jujube germplasm.

**Figure 3 plants-12-00858-f003:**
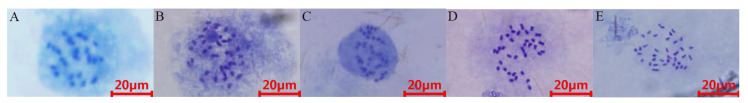
Images of natural triploids and diploid of jujube. (**A**) Linyilizao (diploid), (**B**) QXPGZ; (**C**) TGDGHPZ; (**D**) JSNYDBZ; (**E**) LXEBCSZ.

**Figure 4 plants-12-00858-f004:**
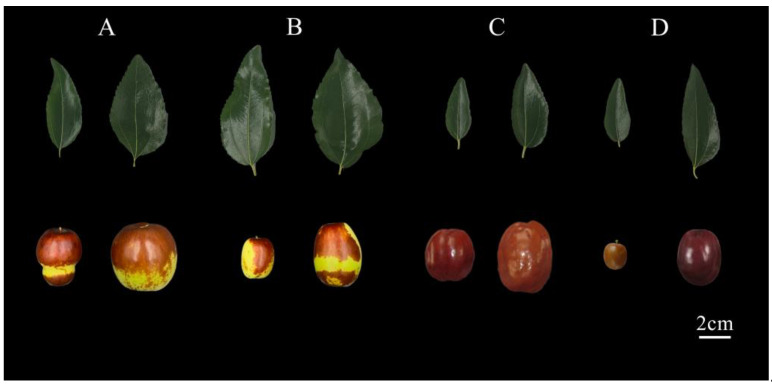
Images of leaves and fruit traits in natural triploids and diploids of jujube. (**A**, right) QXPGZ, (**A**, left) QXMGZ (jujube diploid control in the sub-region). (**B**, right) TGDGHPZ, (**B**, left) TGYZGZ (jujube diploid control in the sub-region). (**C**, right) JSNYDBZ, (**C**, left) JSNYXBZ (jujube diploid control in the sub-region). (**D**, right) LXEBCSZ, (**D**, left) LXEBCCMZ (jujube diploid control in the sub-region.).

**Figure 5 plants-12-00858-f005:**
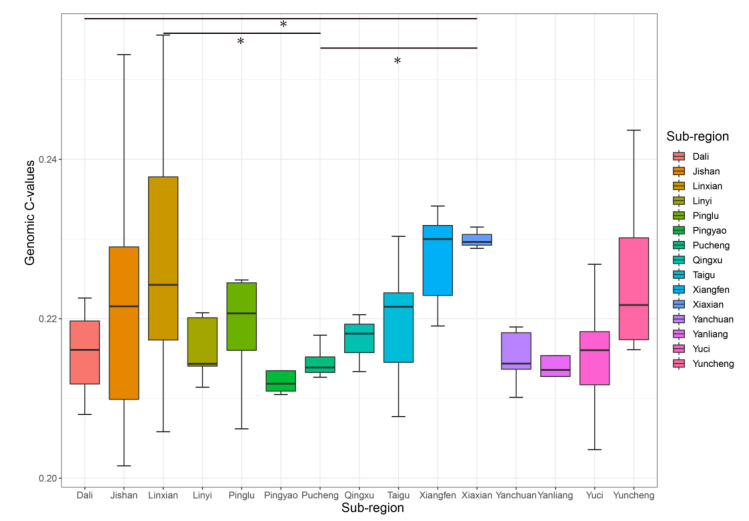
Degree of variation in genomic C-values of jujube germplasm in different sub-regions. Three or more germplasm were selected for analysis. * represents a difference between the two sub-regions at the 0.05 level.

**Figure 6 plants-12-00858-f006:**
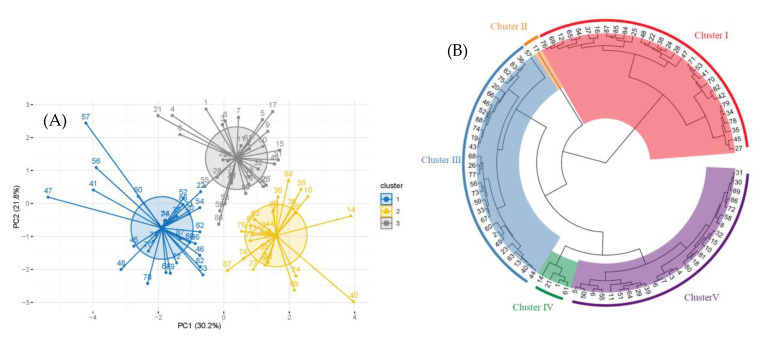
(**A**) Principal component analysis (PCA) of phenotypic traits and genomic C-values of jujube in the YRB of the Shanxi–Shaanxi region. PC1 and PC2 represent PCA contribution. Each serial number represents a species, with gray, blue, and yellow representing the three clusters into which it is divided. See S3 (Appendix A) for serial number details. (**B**) Clustering analysis of phenotypic traits and genomic C-values of jujube in the YRB of the Shanxi–Shaanxi region. See S3 for serial number details. Different colors represent different taxa, and clusters II and IV are jujube germplasms of the Linxian region.

**Figure 7 plants-12-00858-f007:**
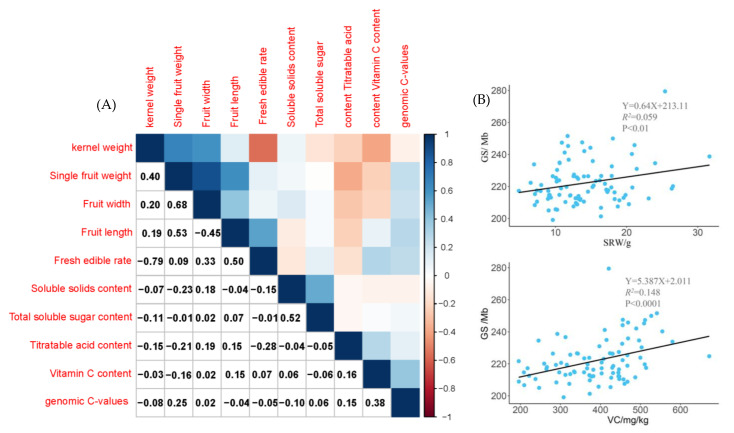
Correlation analysis and linear regression fitting between genomic C-values and fruit traits. (**A**) The negative correlation is shown in blue, positive correlation in orange; the darker the color, the higher the correlation coefficient. (**B**) The coefficient of determination (square of Pearson correlation coefficient) between genomic C-values and single-fruit weight was 0.059 and vitamin C content was 0.148.

**Table 1 plants-12-00858-t001:** Analysis of least significant differences (LSDs) in the genomic C-values of jujube in Linxian County.

No.	Germplasm	Mean(pg) ± SD	LSD Group	No.	Germplasm	Mean ± SD (pg)	LSD Group
1	LXMJW-1	0.25559 ± 0.00455	a	27	LXWWZ	0.21163 ± 0.01031	nop
2	LXMJW-2	0.24662 ± 0.00335	abcde	28	LXYLLZ	0.20582 ± 0.00153	p
3	LXMJW-3	0.24119 ± 0.0097	abcdefg	29	LXLJPAAZ	0.24571 ± 0.00771	abcdef
4	LXMJW-4	0.25135 ± 0.00137	ab	30	LXHGZ	0.21552 ± 0.00298	klmnop
5	LXXZZ-2	0.24856 ± 0.0085	abcd	31	LXGJJZ	0.23797 ± 0.00104	abcdefgh
6	LXXZZ-1	0.25082 ± 0.00826	ab	32	LXXCZ	0.23099 ± 0.00449	defghijkl
7	LXWJZDZ	0.24969 ± 0.01293	abc	33	LXQJSTZ	0.21403 ± 0.00069	lmnop
8	LXYJZ-1	0.25313 ± 0.00008	ab	34	LXYJMYZ	0.23151 ± 0.00345	cdefghijkl
9	LXYJZ-2	0.2378 ± 0.00069	abcdefghi	35	LXLJZ	0.21962 ± 0.00232	ijklmnop
10	LXYJZ-3	0.23711 ± 0.00918	bcdefghij	36	LXQJSLJZ	0.21703 ± 0.00684	klmnop
11	LXTHDMZ-1	0.24027 ± 0.00215	abcdefgh	37	LXQJSLJZ	0.21935 ± 0.00071	jklmnop
12	LXWJZMYZ	0.2273 ± 0.00185	ghijklmno	38	LXQJSDMZ	0.22379 ± 0.00643	ghijklmnop
13	LXSJLNNZ	0.22777 ± 0.00141	fghijklmno	39	LXQJSFZ-2	0.21436 ± 0.00656	lmnop
14	LXQYNNZ	0.21633 ± 0.00311	klmnop	40	LMX1	0.22396 ± 0.00239	ghijklmnop
15	LXCNNZ	0.22209 ± 0.0121	hijklmnop	41	LXSEZ	0.22238 ± 0.00664	hijklmnop
16	LXXZZ-2	0.21841 ± 0.0074	klmnop	42	LXRHZ	0.22202 ± 0.00831	hijklmnop
17	LXYZCZ	0.21886 ± 0.00524	jklmnop	43	LXYZ	0.21734 ± 0.00234	klmnop
18	LXKYDSZ	0.23908 ± 0.0003	abcdefgh	44	LXMZ	0.21845 ± 0.00663	klmnop
19	LXEBCSZ	0.2338 ± 0.00265	bcdefghijk	45	LXDSZ	0.22215 ± 0.0004	hijklmnop
20	LXEBCCMZ	0.23048 ± 0.00431	defghijklm	46	LXSTZ	0.21212 ± 0.00365	nop
21	LXLJHPBB	0.24204 ± 0.00591	abcdefg	47	LXHTZ	0.23116 ± 0.00672	defghijkl
22	SXLXZ22	0.22982 ± 0.0068	efghijklmn	48	LXHZ	0.22509 ± 0.0064	ghijklmno
23	SXLXZ23	0.2122 ± 0.00666	mnop	49	LXDLLZ	0.22425 ± 0.00325	ghijklmno
24	SXLXZ24	0.2144 ± 0.0007	klmnop	50	LXHHZ	0.21893 ± 0.00281	jklmnop
25	SXLXZ25	0.21333 ± 0.00527	lmnop	51	LYHDZ	0.20962 ± 0.00484	op
26	SXLXZ26	0.21418 ± 0.00299	lmnop	52	LXMZLMZ	0.22536 ± 0.00646	ghijklmno
				53	LXMALSZ	0.23322 ± 0.00405	bcdefghijk

Note: F = 14.89, *p* < 0.0001 Significant difference between lowercase letters.

**Table 2 plants-12-00858-t002:** Quantitative trait performance of three clusters of jujube germplasm in the middle Yellow River basin.

Cluster	SRW/g	FLD/cm	FTP/cm	SW/g	SF/%
cluster1	9.21 ± 2.01 C	3.17 ± 0.41 B	2.48 ± 0.23 C	0.49 ± 0.16 B	29.72 ± 2.66 A
cluster2	13.4 ± 2.93 B	3.99 ± 0.39 A	2.71 ± 0.27 B	0.4 ± 0.11 C	30.17 ± 2.81 A
cluster3	19.25 ± 4.26 A	3.88 ± 0.54 A	3.2 ± 0.28 A	0.81 ± 0.19 A	30.66 ± 3.93 A
cluster	TS/%	TA/%	VC/mg/g	FER/%	GS/pg
cluster1	24.96 ± 2.53 b	0.75 ± 0.34 A	381.92 ± 83.48 B	94.53 ± 1.71 B	0.221 ± 0.007 B
cluster2	26.82 ± 3.23 a	0.62 ± 0.19 B	456.48 ± 72.41 A	96.92 ± 0.96 A	0.23 ± 0.01 A
cluster3	25.52 ± 3.11 ab	0.51 ± 0.15 B	322.05 ± 79.85 C	95.1 ± 1.44 B	0.225 ± 0.01 B

Means with the same lowercase and capital letters were not significantly different at *p* < 0.05 and < 0.01, respectively. SRW: Single-fruit weight; FLD: Fruit length; FTP: Fruit width; SW: Stone weight; SF: Soluble solids content; TS: Total soluble sugar content; TA: Titratable acid content; VC: Vitamin C content; FER: fresh edible rate; GS: genomic C-value.

## Data Availability

Data are contained within the article and the Appendix A.

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
