# Peer review of "Genomic C-Value Variation Analysis in Jujube (Ziziphus jujuba Mill.) in the Middle Yellow River Basin"

_plants, 2023, doi:10.3390/plants12040858_

Round 1
Reviewer 1 Report
Comments to the Author
In this manuscript, the authors have estimated the genomic C-values of jujube germplasms in the Yellow River Basin of the Shanxi–Shanxi region and evaluated their differences in different sub-regions. They also analyzed the correlations between the genomic C-values and fruit agronomic traits. Obviously, the large amounts of new data are the biggest light spot of this study, which is definitely helpful for understanding the genetic diversity-related studies of jujube germplasm. However, the presentation, analysis and interpretation of the results are insufficient, and the introduction and discussion do not delve in depth. Therefore, to make this story fully described, some clarifications and conclusions are still needed. Below are some specific suggestions, questions, comments and points that need consideration.
Introduction:
1. In the introduction, the authors have described the history and the diversity of germplasm of jujube, however, they did not raise any questions or explain the main purpose of this manuscript.
2. In the 2nd paragraph of the introduction, they have described the definition and the database of C-value, the method for detecting C-values and the reasons for why the genome size varies, which, to my understand, are not very informative for this study. I, therefore, suggest the authors to summarize the significance and applications of C-value estimation in the germplasm evaluation, following examples of previous studies focusing on the similar subject, if necessary.
3. Also, I suggest the author to explain the purpose of analyzing the correlation of genome size and important fruit traits and lay out main content of this study in the introduction.
Results:
4. In 67, the authors used the “Shanxi Duli”, a Rosaceae species far from jujube, as the internal standard. To my knowledge, the genomes size of several cultivated and wild type jujube, like “Dongzao”, “Junzao” and “Suanzao”, have been conformed based on the genome sequencing. I am wondering why they don’t use these closely related jujube varieties as internal standard.
5. In line 70, what is the unit of the 444.36? Mb or pg?
6. I recommend the authors to give a general description of the sampling strategy before depict the results in the 2nd paragraph of section 1.
7. If it is possible, as one of the main points in this study, I suggest the authors to show the variety of C-value in Figs (maybe point diagram or histogram) or tables in the main text instead of in the supplementary data, as an image is worth more than 1000 words!
8. Without any illustration, the author “speculated that QXPGZ, TGDGHPZ, JSNYDBZ, and LXEBCSZ were triploid” (line 86-87). Could they explain more?
9. I recommend the authors to put an image of diploid jujube in Fig 2 as a control.
10. In Fig 3, the characters in the histograms are too small to read. In addition, as the authors have analyzed the significant difference, I recommend they show the results in the figure 4, which may help to make the conclusion clearer.
11. The authors have compared the mean genomic C-values among different regions in line 93-98, however, what does these results mean? An in-depth conclusion of these results is needed.
12. I am confused about the results described in line 126-140. First of all, I don’t think fig 4 is readable without more illustrations, could they explain more in figure legends? Secondly, I haven’t seen any descriptions about phenotypic traits. So, I am wondering how they reached the conclusions that “different clusters are associated with different traits” just based on the PCA of C-values. Accordingly, I think the interpretation and analysis for the results are insufficient, and the conclusion seems not strong enough.
13. Why do they choose 89 representatives? What is the standard for these representatives?
14. The conclusions in line 142-147 are similar with that in line 155-160, which is verbose. More importantly, the logic of the context is break by other content, which is not easy for understanding. Similar problems can be found in other sections, such as in line 98-100 and 108-110, please check.
15. Why did they analyze the correlation among traits? What did these results explain?
16. According to the results, the single fruit weight is negatively correlated with the Vc content, but the C-values are positively correlated with both of them, how to explain this?
Discussion:
17. In the first and third sections of discussion, the authors have mentioned some previous findings, however, without summarization and comparation. To highlight their findings, an summarize of this study and in-depth comparison of the results with those of the previous studies are needed.
18. In the second section of the Discussion, the authors have mentioned that “older plant lineages have relatively small genomes. Genomic C-values of 0.2191pg for wild jujube and 0.2248pg for Chinese jujube determined in this study suggest that wild jujube may be an ancient species and that the Chinese jujube may have evolved slowly from it.” I am wondering how they understand the C-value paradox?
Author Response
Response to Reviewer 1 Comments
thank you very for your comments and suggestions. those comments are all valuable and very helpful for revising and improving our paper, as well as the important guiding significance to our researches. we have studied comments carefully and have made correction which we hope meet with approval. the main corrections in the paper and the responds to reviewer's comments are as flowing.
responds to the reviewer's comments
Point 1: In the introduction, the authors have described the history and the diversity of germplasm of jujube, however, they did not raise any questions or explain the main purpose of this manuscript.
Response 1: In lines 43-46, the question and the primary purpose have been added.
Point 2: In the 2nd paragraph of the introduction, they have described the definition and the database of C-value, the method for detecting C-values and the reasons for why the genome size varies, which, to my understand, are not very informative for this study. I, therefore, suggest the authors to summarize the significance and applications of C-value estimation in the germplasm evaluation, following examples of previous studies focusing on the similar subject, if necessary.
Response 2: Thank you for your comments. The C-value database was not very useful in this study and has been removed. Because C-value and phenotypic trait correlation analysis is the key to germplasm evaluation, I combined germplasm evaluation, genomic C-value, and phenotypic trait correlation to ultimately describe this study's primary purpose(in lines 57-68).
Point 3: Also, I suggest the author to explain the purpose of analyzing the correlation of genome size and important fruit traits and lay out main content of this study in the introduction.
Response 3: Your comments have been constructive. In lines 58-68, I have revised the article to incorporate your comments.
Point 4: In 67, the authors used the “Shanxi Duli”, a Rosaceae species far from jujube, as the internal standard. To my knowledge, the genomes size of several cultivated and wild type jujube, like “Dongzao”, “Junzao” and “Suanzao”, have been conformed based on the genome sequencing. I am wondering why they don’t use these closely related jujube varieties as internal standard.
Response 4:Flow cytometry works by determining the fluorescence intensity of an unknown genome from the fluorescence intensity of a known genome to calculate the genomic C value. If cultivated and wild jujube are used as internal standards, the peak areas of the fluorescence intensities will likely overlap, and thus the results will be incorrect. However, the genome size of the internal standard should not exceed the genome size of the variety to be tested by too much. For example, we chose wheat (17Gb) as the internal standard to calculate the genome size of jujube germplasm. However, as the genome size difference between the two species is too large, the peak area of fluorescence intensity is too far away, which is likely to lead to incorrect results. Hence, we chose “Shanxi Duli” as the internal standard, one that it and jujube respond to the same degree of woody plant buffer, and the other is that the genome size difference is not large.
Point 5: In line 70, what is the unit of the 444.36? Mb or pg?
Response 5: We have modified it to 444.36 Mb.
Point 6: I recommend the authors to give a general description of the sampling strategy before depict the results in the 2nd paragraph of section 1.
Response 6: We also realize that the second paragraph of the first section does not have a concluding sentence, which we have added(in lines 98-99).
Point 7: If it is possible, as one of the main points in this study, I suggest the authors to show the variety of C-value in Figs (maybe point diagram or histogram) or tables in the main text instead of in the supplementary data, as an image is worth more than 1000 words!
Response 7: Because there is a lot of information in the table, it is difficult to reflect the diversity of the C-value in one map. In order to reflect the diversity of the C-value, I made a map so that not only can the geographical location of the collected germplasm but also the diversity of the C-value be reflected.
Point 8:Without any illustration, the author “speculated that QXPGZ, TGDGHPZ, JSNYDBZ, and LXEBCSZ were triploid” (line 86-87). Could they explain more?
Response 8: Your advice is beneficial in identifying the chromosome part. Perhaps we have described this aspect of ploidy identification less, and the identification of natural triploids is described in detail in lines 108-115
Point 9: I recommend the authors to put an image of diploid jujube in Fig 2 as a control.
Response 9: Pictures of jujube diploids have been placed in the text as a control (Figure 2). And to show the difference between diploids and triploids, we added pictures of leaves and fruits (Figure 3) to support our conclusion further.
Point 10: In Fig 3, the characters in the histograms are too small to read. In addition, as the authors have analyzed the significant difference, I recommend they show the results in the figure 4, which may help to make the conclusion clearer.
Response 10: Your suggestions are beneficial for Figure 3. I only did the box line diagram to reflect the degree of variation, but I did not know that the box line diagram can also show the different relationships, and the font size of the box line diagram and the further analysis has been revised (Figure 5).
Point 11: The authors have compared the mean genomic C-values among different regions in line 93-98, however, what does these results mean? An in-depth conclusion of these results is needed.
Response 11: The average genomic C values for different regions have no specific meaning, are just a primary treatment of the data, and have no influence on the results of this paper. Hence, removed.
Point 12: I am confused about the results described in line 126-140. First of all, I don’t think fig 4 is readable without more illustrations, could they explain more in figure legends? Secondly, I haven’t seen any descriptions about phenotypic traits. So, I am wondering how they reached the conclusions that “different clusters are associated with different traits” just based on the PCA of C-values. Accordingly, I think the interpretation and analysis for the results are insufficient, and the conclusion seems not strong enough.
Response 12: Your suggestions are very helpful for the readability of the article. I have annotated all tables and images in the report. Based on the results of the PCA analysis, we put the mean and difference of fruit traits in different clusters into table 2 to explain that different clusters are associated with different traits
Point 13: Why do they choose 89 representatives? What is the standard for these representatives?
Response 13: Because these 89 germplasm are representative of excellent germplasm from various regions, and in the future, these superb germplasm may be introduced as commercial cultivars, we found some valuable correlations that can guide breeding programs.
Point 14: The conclusions in line 142-147 are similar with that in line 155-160, which is verbose. More importantly, the logic of the context is break by other content, which is not easy for understanding. Similar problems can be found in other sections, such as in line 98-100 and 108-110, please check.
Response 14: Your suggestion is very helpful. First, we removed the correlation among phenotypic traits from the text because it only proves the data's reliability. It cannot influence the conclusion. So keep the correlations among phenotypic characteristics in Figure 7 (left). Second, in lines 192-211, we applied correlation analysis and linear regression analysis, and although the conclusions are similar, they express different meanings, one without the other. However, the content is repeated, and we have revised it.
Point 15: Why did they analyze the correlation among traits? What did these results explain?
Response 15: Because the fruit traits were added in the fourth part, a simple correlation between fruit traits was illustrated to convince the reader of the authenticity of the fruit traits. The text is deleted, and Figure 7 (left) is retained.
Point 16: According to the results, the single fruit weight is negatively correlated with the Vc content, but the C-values are positively correlated with both of them, how to explain this?
Response 16: Thank you for your valuable comments on our article. The comment focuses on as aspect that we did not address in the original manuscript. We used Pearson's correlation analysis, which is a simple correlation analysis of two variables that are not related to the third variable. However, this does not address the effect of other variables on the results; therefore, we revised the correlation coefficients using partial correlation analysis and correspondingly revised the linear regression analysis. We used correlation analysis in order to filter out fruit traits that were correlated with genomic C values. Linear regression was then used to fit the genomic C values to the filtered fruit traits, and the fruit trait prediction equations were subsequently obtained.
Point 17: In the first and third sections of discussion, the authors have mentioned some previous findings, however, without summarization and comparation. To highlight their findings, an summarize of this study and in-depth comparison of the results with those of the previous studies are needed.
Response 17: The first and third sections of the discussion have been revised according to the comment.
Point 18: In the second section of the Discussion, the authors have mentioned that “older plant lineages have relatively small genomes. Genomic C-values of 0.2191pg for wild jujube and 0.2248pg for Chinese jujube determined in this study suggest that wild jujube may be an ancient species and that the Chinese jujube may have evolved slowly from it.” I am wondering how they understand the C-value paradox?
Response 18: Thank you for your suggestion. The C-value paradox is the absence of correspondence between genome size and the number of genes. In particular, there is no direct correspondence between the variation of genome size in eukaryotes and the degree of complexity of the organism. In other words, each group of organisms (e.g., amphibians and mammals) has a minimum and maximum variation in genome size. Hawkins uses cotton as an example. Since cotton diverged from a common ancestor, copia retrotransposons have been active in every strain and continuously proliferate. And finally, it was concluded that transposons may have played an important role in plant evolution. Retrotransposons of this type are first transcribed into RNA, and then the RNA is reverse transcribed and becomes DNA inserted into the target site. Thus one transposition is equivalent to one copy-paste of a gene. Liu, M.J. (2006) Chinese jujube: botany and horticulture is a book that shows through fossil excavations that sour jujube is indeed the ancestor of cultivated jujube, and our findings are in line with Hawkins and Liu, M.J. Returning to the C-value paradox, our understanding is that the C-value I understand the C-value paradox to be mainly for groups of organisms (e.g., amphibians, mammals) in which genome C-values are not related to evolution, but narrowed down to a particular genus, genome size and evolution may be related, and this relationship is inextricably linked to transposons. This is my understanding of the C-value paradox.

Reviewer 2 Report
The manuscript described the use of genomic C-value to evaluate the differences of jujube germplasms in the middle Yellow River Basin (YRB) of the Shanxi–Shaanxi region in China. The highest variation was found in Linxian, among 29 subregions. The germplasms from Linxian were separated from other subregions by cluster analysis. In addition, the genomic C-value was correlated with several fruit agronomic traits, which was a helpful tool for selecting and breeding varieties. Though 186 jujube germplasms were analyzed, the manuscript is not recommended for publication for the following reasons.
Major:
1. Data insufficiency for evaluation:
1.1 There are 29 subregions analyzed; however, only the differences among the jujube germplasm of Linxian County were shown (F = 14.89, p < 0.0001). There are only 15 subregions presented in Figure 3.
1.2 The geographical map of the 29 subregions in the middle Yellow River Basin (YRB) of the Shanxi–Shaanxi region in China was not provided to know their locations.
1.3 Four jujube germplasms were found to be triploid; however, there were no further descriptions to compare their differences with other jujube germplasms, which were diploid—for example, morphology and fruit traits. In addition, control of jujube germplasm, which was diploid, was not shown in Figure 2 for comparison.
1.4 It is impossible to judge if the F-test of ANOVA was performed for the data shown in Table 3. If it is, it should be indicated.
1.5 There needed to be a clue to show why 89 representative jujube germplasm were further selected and analyzed by principal component analysis and cluster analysis. The data shown in Figure 4 needs to be more convincing.
2. Lack of data interpretation:
It showed only the titles for the figures and the tables. Additional legends and footnotes are required for data interpretation.
3. Inappropriate data presentation:
3.1 The definitions of PC1 and PC2 for the principal component analysis were unknown (Figure 4).
3.2 Three terms, taxa, group, and cluster, were mix-used in the text. (Section 2.2).
3.3 Figure 5 and Figure 6 were related. They should be combined into one.
3.4 Figure 5 and Table S1 were the same, but the values differed.
3.5 The GS (Mb) values shown in Table S3 were half of the diploid.
Author Response
Response to Reviewer 2 Comments
thank you very for your comments and suggestions. those comments are all valuable and very helpful for revising and improving our paper, as well as the important guiding significance to our researches. we have studied comments carefully and have made correction which we hope meet with approval. the main corrections in the paper and the responds to reviewer's comments are as flowing.
responds to the reviewer's comments
Point 1: There are 29 subregions analyzed; however, only the differences among the jujube germplasm of Linxian County were shown (F = 14.89, p < 0.0001). There are only 15 subregions presented in Figure 3.
Response 1: Because Linxian germplasm had the largest coefficient of variation, the analysis of variance (ANOVA) was conducted separately for Linxian germplasm. Other subregions were also subjected to ANOVA, and I have added additional data(Table S4).
Point 2: The geographical map of the 29 subregions in the middle Yellow River Basin (YRB) of the Shanxi–Shaanxi region in China was not provided to know their locations.
Response 2: The map of the middle Yellow River Basin (YRB) of the Shanxi–Shaanxi region has been added to Figure 1.
Point 3: Four jujube germplasms were found to be triploid; however, there were no further descriptions to compare their differences with other jujube germplasms, which were diploid—for example, morphology and fruit traits. In addition, control of jujube germplasm, which was diploid, was not shown in Figure 2 for comparison.
Response 3: Diploid jujube germplasm chromosome controls have been added. A comparison of diploid and triploid leaf and fruit traits has been added. And I think the process of determining triploidy is not described clearly, and we add to the words of the article(lines 108-127).
Point 4: It is impossible to judge if the F-test of ANOVA was performed for the data shown in Table 3. If it is, it should be indicated
Response 4: I assume you are referring to Table 1; we have added the F value at the bottom of the table.
Point 5: There needed to be a clue to show why 89 representative jujube germplasm were further selected and analyzed by principal component analysis and cluster analysis. The data shown in Figure 4 needs to be more convincing.
Response 5: Principal component and cluster analysis are widely used in plant taxonomy. and explained in lines 252 - 255.
Point 6: It showed only the titles for the figures and the tables. Additional legends and footnotes are required for data interpretation.
Response 6: Legend and footnotes have been added to the tables and images in the text.
Point 7: The definitions of PC1 and PC2 for the principal component analysis were unknown (Figure 4).
Response 7: PC1 and PC2 represent the contribution rates in the principal component analysis, which we have explained in the figure notes of Figure 6.
Point 8: Three terms, taxa, group, and cluster, were mix-used in the text. (Section 2.2).
Response 8: The confusing use of taxa, group, and cluster has led to confusion in the article, and the text has been unified as cluster.
Point 9: Figure 5 and Figure 6 were related. They should be combined into one.
Response 9: We have combined the correlation analysis chart and the linear regression chart.
Point 10: Figure 5 and Table S1 were the same, but the values differed.
Response 10: The correlation coefficients in Figure 7 and Table S1 are different, probably due to the different algorithms applied by R and SPSS. It has been harmonized. And, based on the suggestion of another reviewer, we applied partial correlation analysis, which allowed us to make the results more confident.
Point 11: The GS (Mb) values shown in Table S3 were half of the diploid.
Response 11: In Table S3, the genome size (Mb) is indeed the diploid genome size, not the haploid genome size. Perhaps the expert meant that the haploid genome size should be shown because, typically, the reader knows the diploid genome size and not the haploid genome size, so we show the diploid genome size and show the ploidy in ploidy for the reader's convenience.

Reviewer 3 Report
The manuscript presents analysis of C-values in 186 accessions in Ziziphus jujuba Mill.. The authors also studied correlation of various phenotypical traits with the corresponding C-values. Interestingly, the authors have found correlation between C-values and the ascorbic acid content in fruits and some othe agronomically important traits. All these data are original and they can be useful both for breeders and for the scientists studying Ziziphus jujube. It appears, however, that in its current form, the manuscript is focused on very narrow spectrum of readers. There are also no attempts to biologically explain some observed phenomena. It is hard to explain presence of triploidy in natural population. Possibly, it could be explained by the vegetative propagation of such clones. Alternatively, they could be a product of the recent cross of tetraploid with diploid. It is also not clear how much is correlation of ascorbic acid content with C-values caused by the presence of triploids in the dataset. Minor intraspecific C-value variation (not so pronounced as that caused by changes in ploidy) is often caused by variation in the content of various repetitive sequences in the genome. Previous studies in Ziziphus jujuba Mill. Have shown that there is a correlation with higher ploidy and ascorbic acid content, these finding has an easy biological interpretation as the change in ploidy is supposed to cause changes in gene expression. On the other hand, the changes in repetitive elements can show random effects on gene expression (e.g. due to effect of transposons and retrotransposons) and so the correlation could be rather connected with the origin of ascorbic acid rich accessions from the ancestor with higher genome size. Alternatively, endopolyploidy could be taken into account. Unfortunately, the study of genome size itself is probably not sufficient for the explanation of the system from the biological point of view. A significant improvement could be, however, done by the study of the enrichment of various classes of repeats based on low coverage sequencing (using RepeatExplorer or similar software) of several representative accessions chosen from each cluster.
Author Response
Response to Reviewer 3 Comments
Thank you very much for your comments and suggestions. All comments are valuable and very helpful for revising and improving our manuscript, as well as elevating the significance to our research. We have comprehensively reviewed the comments and revised the manuscript accordingly and we hope your approval requirements have been met, and through your guidance of our article can be published successfully. The revisions have been incorporated in the manuscript and individual responses to reviewer's comments are outlined below.
Point :The manuscript presents analysis of C-values in 186 accessions in Ziziphus jujuba Mill.. The authors also studied correlation of various phenotypical traits with the corresponding C-values. Interestingly, the authors have found correlation between C-values and the ascorbic acid content in fruits and some other agronomically important traits. All these data are original and they can be useful both for breeders and for the scientists studying Ziziphus jujube. It appears, however, that in its current form, the manuscript is focused on very narrow spectrum of readers. There are also no attempts to biologically explain some observed phenomena. It is hard to explain presence of triploidy in natural population. Possibly, it could be explained by the vegetative propagation of such clones. Alternatively, they could be a product of the recent cross of tetraploid with diploid. It is also not clear how much is correlation of ascorbic acid content with C-values caused by the presence of triploids in the dataset. Minor intraspecific C-value variation (not so pronounced as that caused by changes in ploidy) is often caused by variation in the content of various repetitive sequences in the genome. Previous studies in Ziziphus jujuba Mill. Have shown that there is a correlation with higher ploidy and ascorbic acid content, these finding has an easy biological interpretation as the change in ploidy is supposed to cause changes in gene expression. On the other hand, the changes in repetitive elements can show random effects on gene expression (e.g. due to effect of transposons and retrotransposons) and so the correlation could be rather connected with the origin of ascorbic acid rich accessions from the ancestor with higher genome size. Alternatively, endopolyploidy could be taken into account. Unfortunately, the study of genome size itself is probably not sufficient for the explanation of the system from the biological point of view. A significant improvement could be, however, done by the study of the enrichment of various classes of repeats based on low coverage sequencing (using RepeatExplorer or similar software) of several representative accessions chosen from each cluster.
Response 1: We determined that the four germplasms were triploids by flow cytometry (Table S3) and chromosome counting (Figure 3). Moreover, the triploids were compared with diploids for phenotypic traits (Figure 4). Figure 4 illustrates the phenotypic trait differences in diploid and triploid. In most fruit trees the most commonly used crosses for breeding are facing serious obstacles in jujube trees (extremely small flowers that are difficult to remove, low fruit set rate of only about 1%, and high failure rate.) According to previous studies, natural triploidy does exist in dates, and it is suggested that natural triploidy in dates may be the result of natural crosses between unminimized gametes of diploid varieties and normal gametes of the same or different varieties [1]. The test materials selected for this study, all from wild jujube varieties in sub-regions, can be identified as natural triploids. Relevant reference has been cited to support this: [1] Pang, j.y.; Liu, P.; Zhou, J.Y.; Peng, S.Q.; Cao, Q.G.; Chu, X.F. Karyotypes of different strains in Ziziphus jujuba Mill, 'Zanhuang Dazao'. Acta Horticulturae Sinica. 2005, 32(5): 798–801.
Response 2: In lines 274-275, to exclude the effect of ploidy on correlation analysis, we selected germplasm that was diploid and used the DNA content of haploid for the analysis (C-values).
Response 3: We thank the reviewers for the suggestions, which have significantly improved our manuscript. This study focuses on Genomic C-value Variation Analysis in Jujube (Ziziphus jujuba Mill.) in the Middle Yellow River Basin and is a phased progress in the correlation of fruit traits with genomic C-values. However, fruit traits and genomic C values were not linked with the intention to focus on a limited spectra of readers. Therefore, we reviewed the effect of transposable elements on phenotypic traits in the discussion section and delved into the relationship between genomic C-values and fruit traits. We modified the discussion section as follows: "Genome size varies significantly between and within species, and the causes of variation may be recombination rates, tandem repeats, proliferation of transposable elements, and ecological factors [48], while transposable elements (TEs) may have played an important role in plant evolution [42]. Insertion of TEs affects the expression of host genes in several ways. Domínguez et al [49] found TEs insertions within or near 1 kb of genes and compared them to the same genes without TEs insertions at the transcriptional level and found that most of the genes with TEs insertions were expressed more than twice those without them. If TEs are inserted into the cis-regulatory region, they may enhance gene expression and produce phenotypic differences. For example, studies on red grapes, white grapes, and blood oranges all found that insertion of TEs into the upstream promoters of genes can cause phenotypic alterations [50–51]. It is important to note that the distribution of TEs in the host genome is not random and they show a strong preference for specific genomic regions, a preference that both reduces damage to the host and maximizes an opportunity of TEs expression [52–53]. In this study, genomic C-values were found to be positively correlated with vitamin C content and single fruit weight, probably due to the insertion of transposable elements, resulting in increased expression of the relevant genes. In-depth investigations of the relationship between genomic C-values and phenotypic traits will be conducted, using genomic enriched repeat sequences,” (lines 302–320).
Response 4: Thank you very much for the valuable comments. We have combined ploidy studies with C-value diversity, to highlight the topic while keeping the manuscript compact. We have revised the discussion section as follows: "Previous studies have illustrated that plant polyploidy promotes genome evolution as well as species diversity," (line 271-272).

Round 2
Reviewer 2 Report
The revised manuscript described the use of genomic C-value to evaluate the differences of 186 jujube germplasms in the middle Yellow River Basin (YRB) of the Shanxi–Shaanxi region in China. The highest variation was found in Linxian, among 29 subregions. The germplasms from Linxian were separated from other subregions by cluster analysis. In addition, the genomic C-value was correlated with several fruit agronomic traits, which may be a helpful tool for selecting and breeding varieties. Though the manuscript was improved, it is not recommended for publication in Plants because the critical questions were not answered. The primary reasons are listed as follows.
Major:
1. The geographical map of the 29 subregions in the middle Yellow River Basin (YRB) of the Shanxi–Shaanxi region in China was already provided (Figure 2). However, they should be labeled by numbers. The locations of each number should be described in a supplementary table. The readers would like to know the GPS coordinates of each location for geographical relationships.
2. Four jujube germplasms were found to be triploid. An additional Figure 4 was provided to show the leaf morphology and fruit traits. However, the diploid controls for (A)-(D) in the subregions should be described because they are different.
3. The reasons 89 representative jujube germplasm were further selected and analyzed by principal component analysis and cluster analysis from 186 jujube germplasms were not explained.
4. The definitions of PC1 and PC2 for the principal component analysis needed to be explained (Figure 6). The authors described that “PC1 and PC2 represent the contribution of principal component analysis” in the legend. However, the readers would like to know what factor was defined as PC1 and which was defined as PC2. For example, they were genome sizes and phenotypic traits, respectively.
5. The authors explained that “the correlation coefficients in Figure 7 and Table S1 are different, probably due to the different algorithms applied by R and SPSS. It has been harmonized.” It would be necessary to explain why the harmonization should be performed and how to do it.
6. Linear regression between genomic C-values and fruit traits, such as FLD, FTP, and FER, shown in the original Figure 6, were deleted. Only two, SRW and VC, were shown in the revised Figure 7. Why?
7. The GS (Mb) values shown in Table S2, not Table S3, were only half of the diploid. The diploid genome size should be shown as the response 11 answered by the authors.
8. Because there were only 15 subregions presented in Figure 5, the authors provide Table S4 to show the genomic C-values of 22 subregions except for Linxian county. However, the total subregions are 29. It is suggested to show all the data of 29 subregions in Table S4.
Author Response
Response to Reviewer 2 Comments
Thank you very much for your comments and suggestions. All comments are valuable and very helpful for revising and improving our manuscript, as well as elevating the significance to our research. We have comprehensively reviewed the comments and revised the manuscript accordingly and we hope your approval requirements have been met, and through your guidance of our article can be published successfully. The revisions have been incorporated in the manuscript and individual responses to reviewer's comments are outlined below.
Point 1: The geographical map of the 29 subregions in the middle Yellow River Basin (YRB) of the Shanxi–Shaanxi region in China was already provided (Figure 2). However, they should be labeled by numbers. The locations of each number should be described in a supplementary table. The readers would like to know the GPS coordinates of each location for geographical relationships.
Response 1: Thank you for your comments, which will improve the manuscript readability. The coordinates of each germplasm have been added in Table S3.
Point 2: Four jujube germplasms were found to be triploid. An additional Figure 4 was provided to show the leaf morphology and fruit traits. However, the diploid controls for (A)-(D) in the subregions should be described because they are different.
Response 2:The suggestions have significantly enhanced our manuscript. A more detailed caption has been added to Figure 4. The names of the germplasm controls have been provided, for example, QXMGZ (jujube diploid control in the sub-region).
Point3: The reasons 89 representative jujube germplasm were further selected and analyzed by principal component analysis and cluster analysis from 186 jujube germplasms were not explained.
Response 3:Thank you for the helpful for the article. The 89 germplasms are representatives of excellent germplasms from various regions, which may also be introduced as commercial cultivars in the future. We found some valuable correlations that can guide breeding programs through these germplasms. Section 2.3 of the manuscript has been modified as follows: “To investigate the association between genome size and phenotypic traits, 89 excellent germplasm representatives from different sub-regions89 representative germplasms were selected for correlation analysis., in 194-195 line.”
Point 4:The definitions of PC1 and PC2 for the principal component analysis needed to be explained (Figure 6). The authors described that “PC1 and PC2 represent the contribution of principal component analysis” in the legend. However, the readers would like to know what factor was defined as PC1 and which was defined as PC2. For example, they were genome sizes and phenotypic traits, respectively.
Response 4:We appreciate the helpful comments. Tables E1 and E2 outline the total variance and component matrix in the principal component analysis. From Table E1, the first two principal components correspond to PC1 and PC2 in Figure 6, and percentage variance (%) corresponds to the contribution of principal components in Figure 6. PC1 and PC2 in Table E2 correspond to PC1 and PC2 in Figure 6. All traits are included in PC1 and PC2 on Table E2, however, the single factor defined as PC1 or PC2 cannot be indicated. Overall, PC1 and PC2 in Figure 6 are the explanatory rates for the traits (30.2% + 21.8%). Related literature has been cited to support this: [1] Sarikhani, S.; Arzani, K.; Karimzadeh, G.; Shojaeiyan, A.; Ligterink, W. Genome Size: A Novel Predictor of Nut Weight and Nut Size of Walnut Trees. HORTSCIENCE. 2018, 53, 275-282. [2] Kadkhodaei, S.; Arzani, K.; Yadollahi A.; Karimzadeh, G.; Abdollahi, H. Genetic Diversity and Similarity of Asian and European Pears (Pyrus Spp.) Revealed by Genome Size and Morphological Traits Prediction. Int j fruit sci. 2021, 21, 619–633.
表E1 Principal component analysis explains the total variance
|
Component |
Initial eigenvalues |
Extraction sums of squared loadings |
||||
|
|
Total |
Percentage of |
Cumulative percentage ( % ) |
Total |
Percentage of |
Cumulative percentage ( % ) |
|
PC1 |
3.021 |
30.211 |
30.211 |
3.021 |
30.211 |
30.211 |
|
PC2 |
2.186 |
21.86 |
52.071 |
2.186 |
21.86 |
52.071 |
|
PC3 |
1.524 |
15.237 |
67.308 |
1.524 |
15.237 |
67.308 |
|
PC4 |
1.185 |
11.854 |
79.162 |
1.185 |
11.854 |
79.162 |
|
PC5 |
0.6 |
6.004 |
85.166 |
|
|
|
|
PC6 |
0.541 |
5.409 |
90.575 |
|
|
|
|
PC7 |
0.452 |
4.523 |
95.098 |
|
|
|
|
PC8 |
0.347 |
3.468 |
98.566 |
|
|
|
|
PC9 |
0.08 |
0.801 |
99.368 |
|
|
|
|
PC10 |
0.063 |
0.632 |
100 |
|
|
|
表E2 Principal component analysis component matrix
|
trait |
Component |
|||
|
PC1 |
PC2 |
PC3 |
PC4 |
|
|
SRW/g |
0.963 |
0.11 |
-0.05 |
0.095 |
|
FLD/cm |
0.588 |
0.599 |
-0.084 |
-0.155 |
|
FTP/cm |
0.888 |
0.043 |
0.053 |
0.186 |
|
SW/g |
0.733 |
-0.528 |
-0.143 |
0.271 |
|
SF/% |
0.113 |
-0.187 |
0.847 |
0.149 |
|
TS/% |
0.022 |
0.131 |
0.873 |
-0.01 |
|
TA/% |
-0.499 |
0.031 |
-0.082 |
0.643 |
|
VC/mg/(100g) |
-0.357 |
0.62 |
-0.01 |
0.385 |
|
FER/% |
0.043 |
0.85 |
0.058 |
-0.406 |
|
genome size/Mb |
0.175 |
0.612 |
-0.031 |
0.544 |
Point 5: The authors explained that “the correlation coefficients in Figure 7 and Table S1 are different, probably due to the different algorithms applied by R and SPSS. It has been harmonized.” It would be necessary to explain why the harmonization should be performed and how to do it.
Response 5: We apologize for the oversight. As this is the first time we submitted a manuscript to this journal, I thought I would send responses to both Reviewer 1 and Reviewer 2 comments as one, so I did not inform you about this revision. Your suggestions helped a lot with improvement of the manuscript. We used Pearson's correlation analysis, which is a simple correlation analysis of two variables, unrelated to the third variable. However, this does not address the effect of other variables on the results; therefore, we revised the correlation coefficients using partial correlation analysis and correspondingly revised the linear regression analysis. We used correlation analysis in order to filter out fruit traits that were correlated with genomic C-values. Linear regression was then used to fit the genomic C-values to the filtered fruit traits, subsequently obtaining the fruit trait prediction equations. Revision in the manuscript was done by including “It has been harmonized” in the first round, meaning that Figure 7 and Table S1 have been harmonized according to the partial correlation analysis.
Point 6: Linear regression between genomic C-values and fruit traits, such as FLD, FTP, and FER, shown in the original Figure 6, were deleted. Only two, SRW and VC, were shown in the revised Figure 7. Why?
Response 6: Reviewer's comments are appreciated. According to the results of the new correlation analysis, genomic C-values were not correlated with FLD, FTP, and FER. Having correlation between two variables is the basis of linear regression. Therefore, the revised linear regression plot of genomic C-value and FLD, FTP, and FER has been deleted.
Point 7:The GS (Mb) values shown in Table S2, not Table S3, were only half of the diploid. The diploid genome size should be shown as the response 11 answered by the authors.
Response 7: Reviewers' comments enhanced the readability of the manuscript and we appreciate this. Table S2 has been modified to the diploid genome size and the corresponding ploidy has been labeled.
Point 8: Because there were only 15 subregions presented in Figure 5, the authors provide Table S4 to show the genomic C-values of 22 subregions except for Linxian county. However, the total subregions are 29. It is suggested to show all the data of 29 subregions in Table S4.
Response 8: Thank you for the suggestion. It has immensely improved the content of the manuscript. Figure 5 shows the coefficient of variation of jujube germplasm genomic C-values from different sub-regions, and the coefficient of variation analysis was based on three or more data. Therefore, to reflect the authenticity and reliability of the data, we selected sub-regions with at least three germplasm datasets for the degree of variation analysis. The analysis of significant variation in Figure 5 is the analysis of variation between sub-regions, which is different from Table S4. Table S4 is the analysis of significant differences in genomic C-values of jujube germplasm within sub-regions. And the analysis of significant differences was divided into two types, t-test for comparing the differences between two datasets and ANOVA for the differences between three or more datasets. Hence, for the authenticity and reliability of the data, we selected sub-regions with more than two germplasm datasets for the analysis of variance. However, we also realized that we should show all sub-regions, so Table S4 has been modified as suggested.

Round 3
Reviewer 2 Report
The revised manuscript reported the use of genomic C-value to evaluate the differences of 186 jujube germplasms in the middle Yellow River Basin (YRB) of the Shanxi–Shaanxi region in China. The highest variation was found in Linxian, among 29 subregions. The germplasms from Linxian were separated from other subregions by cluster analysis. In addition, the genomic C-value was correlated with several fruit agronomic traits, which may be a helpful tool for selecting and breeding varieties. The revised version addressed and answered all the questions and suggestions. It seems to be suitable for publication in the special issue of the journal Plants. Below are the suggestions for the authors to improve their manuscript before acceptance.
1. Abstract. Use cluster to replace taxon as described in section 2.2 and Figure 6.
2. Figure 3. Five sub-images should be labeled with A-E, respectively.
3. Section 4.1. A total of 186 jujube germplasm resources was not shown in Table S1 but in Table S3. In addition, the date of collection should be indicated.
4. The information on the back matter that provided supplementary materials was wrong. The descriptions of Tables S1-S4 should be corrected.
5. The format of references should follow the rules of the journal Plants.
Author Response
Response to Reviewer 2 Comments
thank you very for your comments and suggestions. those comments are all valuable and very helpful for revising and improving our paper, as well as the important guiding significance to our researches. we have studied comments carefully and have made correction which we hope meet with approval. We hope that under the guidance of the Reviewer, our article can be published successfully. the main corrections in the paper and the responds to reviewer's comments are as flowing.
Point 1: Abstract. Use cluster to replace taxon as described in section 2.2 and Figure 6.
Response 1: Thanks for reviewer's comments, which will help the reader to read. The abstract has been modified according to the recommendations.
Point 2: Figure 3. Five sub-images should be labeled with A-E, respectively.
Response 2:reviewer's suggestions have helped the article a lot. In 120-121 line, Five sub-images have been marked with corresponding letters.
Point3: Section 4.1. A total of 186 jujube germplasm resources was not shown in Table S1 but in Table S3. In addition, the date of collection should be indicated.
Response 3:Reviewer's comments are very helpful for the article. Because of the sequence error of supplementary materials, Table S1 did not show 186 jujube germplasm resources information. The sequence of supplementary materials has been modified in 400-405 line. The collection date of jujube germplasm resources has been added to the note of table S1.
Point 4: The information on the back matter that provided supplementary materials was wrong. The descriptions of Tables S1-S4 should be corrected.
Response 4:Reviewer's comments are very helpful for the article. Tables S1-S4 have been corrected
Point 5: The format of references should follow the rules of the journal Plants.
Response 5:References have been revised according to the rules of the journal Plants.
